# Active learning in undergraduate classroom dental education- a scoping review

**Arnaldo Perez** [1]*, **Jacqueline Green**[1], **Mohammad Moharrami**[2], **Silvia Gianoni-Capenakas**[1], **Maryam Kebbe**[3], **Seema Ganatra**[1], **Geoff Ball**[4], **Nazlee Sharmin**[1]

**1** School of Dentistry, Faculty of Medicine & Dentistry, University of Alberta, Edmonton, Alberta, Canada, **2** Faculty of Dentistry, University of Toronto, Toronto, Ontario, Canada, **3** Faculty of Kinesiology, University of New Brunswick, Fredericton, New Brunswick, Canada, **4** Department of Pediatrics, Faculty of Medicine & Dentistry, University of Alberta, Edmonton, Alberta, Canada

* perezgar@ualberta.ca

## Abstract

### Introduction

Previous reviews on active learning in dental education have not comprehensibly summarized the research activity on this topic as they have largely focused on specific active learning strategies. This scoping review aimed to map the breadth and depth of the research activity on active learning strategies in undergraduate classroom dental education.

### Methods

The review was guided by Arksey & O'Malley's multi-step framework and followed the PRISMA Extension Scoping Reviews guidelines. MEDLINE, ERIC, EMBASE, and Scopus databases were searched from January 2005 to October 2022. Peer-reviewed, primary research articles published in English were selected. Reference lists of relevant studies were verified to improve the search. Two trained researchers independently screened titles, abstracts, and full-texts articles for eligibility and extracted the relevant data.

### Results

In total, 93 studies were included in the review. All studies performed outcome evaluations, including reaction evaluation alone (n = 32; 34.4%), learning evaluation alone (n = 19; 20.4%), and reaction and learning evaluations combined (n = 42; 45.1%). Most studies used quantitative approaches (n = 85; 91.3%), performed post-intervention evaluations (n = 70; 75.3%), and measured student satisfaction (n = 73; 78.5%) and knowledge acquisition (n = 61; 65.6%) using direct and indirect (self-report) measures. Only 4 studies (4.3%) reported faculty data in addition to student data. Flipped learning, group discussion, problem-based learning, and team-based learning were the active learning strategies most frequently evaluated (≥6 studies). Overall, most studies found that active learning improved satisfaction and knowledge acquisition and was superior to traditional lectures based on direct and indirect outcome measures.

**Data Availability Statement:** All relevant data are within the manuscript and its Supporting Information files.

**Funding:** The authors received funding (SDERF-02) for this work from the School of Dentistry at the University of Alberta. SG was the PI. The funder had no role in study design, data collection and analysis, decision to publish, or preparation of the manuscript.

**Competing interests:** The authors have declared that no competing interests exist.

## Conclusion

Active learning has the potential to enhance student learning in undergraduate classroom dental education; however, robust process and outcome evaluation designs are needed to demonstrate its effectiveness in this educational context. Further research is warranted to evaluate the impact of active learning strategies on skill development and behavioral change in order to support the competency-based approach in dental education.

## Introduction

Active learning (AL) has been broadly defined as a type of learning that demands active gathering, processing, and application of information rather than passive assimilation of knowledge [1]. This form of learning is well aligned with principles of adult learning, including self-direction, purposefulness, experience-based, ownership, problem orientation, mentorship, and intrinsic motivation [2]. Because students regularly enroll in dental programs as young adults after completing an undergraduate degree, active learning has been encouraged in dental education to help students gain knowledge and develop basic and advanced dental, cognitive, and social skills [3]. Active learning, along with curricular integration, early exposure to clinical care, and evidence-based teaching and assessment are important reforms introduced in dental education to ensure that students develop the competencies they need to become entry-level general dentists in the 21st century [4].

Numerous teaching strategies have been developed to promote active learning across health professions education, including problem-based learning, case-based learning, flipped learning, team-based learning, and group discussion. Research suggests that students and instructors positively value active learning [5, 6]; however, inconclusive evidence exists on the actual impact of active learning on knowledge acquisition, skill development, and attitudinal change in health sciences education [7, 8].

Many studies have been conducted on active learning in dental education, especially in the last two decades. Some primary and review studies have found that active learning is well received by students and instructors and may be more effective than traditional lecture-based teaching in dental education [9, 10]. However, review studies, in particular, have fallen short of providing a comprehensive overview of the existing literature on active learning in dental education [9, 11, 12]. For example, they have largely focused on the outcomes of a few active learning strategies (e.g., problem-based learning, flipped learning) providing limited data on their implementation and evaluation designs. These review studies have also failed to differentiate the scope, range, and nature of the research activity on active learning in different learning environments, including classroom dental education. This learning environment has unique characteristics and is of particular importance because it provides the foundational knowledge that students are expected to apply in laboratory and clinical settings.

Our scoping review aimed to map the breadth and depth of the research activity on active learning strategies in undergraduate classroom dental education from January 2005 to October 2022. Mapping this extensive body of literature is important to inform future research directions on active learning in dental education.

## Methods

The scoping review framework developed by Arksey & O'Malley (2005) guided the study design, which includes the following stages: (1) formulating research questions, (2) identifying

potentially relevant studies, (3) selecting relevant studies, (4) charting the data, and (5) collating, summarizing, and reporting results [13]. Unlike systematic reviews that typically synthesize the existing evidence on relationships between exposure and outcome variables, scoping reviews are well suited to map the breadth and depth of the research activity on complex topics and identify gaps in the relevant literature [13]. Our review report followed the guidelines of PRISMA Extension for Scoping Reviews [14].

## Stage 1: Formulating research questions

Our scoping review sought to answer the following questions:

- What are the characteristics of the studies conducted on active learning in classroom dental education in the study period?

- How were active learning strategies evaluated?

- What were the main results of the studies conducted?

## Stage 2: Identifying potentially relevant studies

Four databases (MEDLINE, ERIC, EMBASE, and Scopus) were searched from January 2005 to October 2022. A preliminary search suggested that most studies on the study topic were published in the last two decades and the quality of the reports produced had substantially improved in the same study period. The search strategy for MEDLINE was developed by two authors (JG and AP) in consultation with a librarian at the University of Alberta. This strategy was then adapted for each database included in the review. Search terms used in each database are shown in Table 1. Reference lists of included studies and articles selected in previous

**Table 1. Detail of search terms and search results.**

| Database | Search Terms | Search Results (Number of Papers) | Year |
|---|---|---|---|
| MedLine | [active learn. OR problem based learning.mp. or exp Problem-Based Learning/ OR case based learning.mp. OR Group adj2 discuss*).mp. OR (small adj2 group*).mp. OR (small adj2 group*).mp. OR (peer adj2 teach*).mp. OR (critical adj2 think*).mp. OR (role adj2 play*).mp. OR team based learning.mp. OR (peer adj2 learn*).mp. OR (flipped adj2 class*).mp. OR (flipped adj2 learn*).mp. OR (blended adj2 learn*).mp.] AND [class.mp. OR class*.mp. OR classes.mp. OR preclinical.mp. OR non-clinical.mp. OR in-class.mp. OR course.mp. OR courses.mp.] AND [exp Students, Dental/ OR exp Education, Dental/ OR (dental adj2 learn*).mp. OR ((dental or dentist*) adj2 (educat* or learn* or student* or teach* or instruct* or curricul*)).mp. OR exp Schools, Dental/] | 422 | 2005–2022 |
| ERIC | [exp Active Learning/ or active learn*.mp. OR Case based learning.mp. or exp "Case Method (Teaching Technique)"/ OR case-based learning.mp. OR problem based learning.mp. or exp Problem Based Learning/ OR problem-based learning.mp. OR (think* adj1 pair* adj1 share*).mp. OR (peer* adj2 learn*).mp. OR critical adj2 think*).mp. OR exp Critical Thinking/ OR (role adj2 play).mp. OR exp Classrooms/ or class*.mp. OR discuss*.mp. or exp Discussion Groups/ or exp Discussion/ or exp Group Discussion/ OR reflection.mp. or exp Reflection/ OR teaching methods.mp. or exp Teaching Methods/] AND [((dental or dentist*) adj2 (educat* or learn* or student*)).mp. OR undergraduate dent*.mp. OR dental schools.mp. or exp Dental Schools/ OR exp Dentistry/ OR dental college.mp.] | 132 | 2005–2022 |
| Scopus | [active learn* OR Problem based Learn* OR Case based learn* OR Group discuss* OR think pair share OR Peer learn* OR "peer teach* OR critical think* OR Role play* OR flipped learn* OR Flipped Class* OR blended learn*] AND [Class* OR preclinical OR non-clinical OR in-class OR course*] AND [dental school OR dentistry OR dental learn* OR dental educat* OR dental student* OR dental teach* OR dental instruct* OR dental curricul*] | 442 | 2005–2022 |
| EMBASE | [active learn*.mp. OR problem based learning.mp. or exp Problem Based Learning/ OR exp problem based learning/ OR case-based learning.mp. OR (Group adj2 discuss*).mp. OR (small adj2 group*).mp. OR (think* adj1 pair* adj1 share*).mp. OR (peer adj2 learn*).mp. OR (peer adj2 teach*).mp. OR (critical adj2 think*).mp. OR (role adj2 play*).mp. OR team based learning.mp. OR (peer adj2 learn*).mp. OR (flipped adj2 class*).mp. OR (flipped adj2 learn*).mp. OR teaching methods.mp. or exp teaching/ OR (blended adj2 learn*).mp.] AND class*.mp. OR preclinical.mp. OR non-clinical.mp. OR in-class.mp. OR course*.mp.] AND [exp dental student/ OR exp dental education/ OR (dental adj2 learn*).mp. OR ((dental or dentist*) adj2 (educat* or learn* or student* or teach* or instruct* or curricul*)).mp. OR dental school.mp.] | 1200 | 2005–2022 |

reviews on specific active learning strategies were verified to enhance the search and test its sensitivity.

### Stage 3: Selecting relevant studies

Inclusion and exclusion criteria were based on the research questions and refined during the screening process. Primary studies published in English were included if they met the following criteria: (i) focused on undergraduate dental education in classroom settings, (ii) used at least one active learning strategy, (iii) involved dental students, and (iv) reported dental student data when students from other programs (e.g., medical students) were involved in the study. Studies were excluded if they were published in a language other than English, reported active learning in clinical or laboratory settings or at program level, and were not available as full-text articles. Review studies and perspective articles were also excluded. No restrictions were set on research methods. All references were exported to Zotero and duplicates were removed by JG. The remaining papers were then exported to Rayyan. A training session was held to ensure understanding of inclusion and exclusion criteria and consistency in their application. Two researchers (JG and SGC) independently screened for titles and abstracts and three researchers independently reviewed the full texts of articles selected in the first phase of screening (JG, SGC, MM). Consensus was obtained by discussion or consulting a fourth reviewer (AP).

### Stage 4: Charting the data

A piloted, literature-informed data collection form was used to extract data on publication (year of publication, country of publication), study characteristics (research inquiry, research methodology, means of data collection), participant characteristics (type of student, sample size), intervention (content area, active learning strategy, comparator, and length of the exposure), evaluation (type of evaluation, level of evaluation, evaluation design, and outcome of interest) and main findings. Data extraction was completed independently by two trained researchers (JG and MM) and the completed data extraction forms were compared. Consensus was obtained by discussion or consulting a third reviewer (AP). Authors of studies that did not report key aspects included in the data extraction form were contacted to provide that information. Missing information was then categorized as "not reported."

### Stage 5: Collating, summarizing, and reporting results

Descriptive statistics were used to summarize quantifiable data using previously developed or data-driven classifications. Evaluation data such as level, outcomes (directly and indirectly measured), and results were summarized according to Kirkpatrick's Model (1998) [15]. This model suggests four levels of outcome evaluation, including *reaction* (satisfaction and perceived outcomes), *learning* (direct measures of outcomes such knowledge, skills, and attitudes), *behavior* (behavioral changes resulting from the intervention), and *results* (organizational changes resulting from the intervention).[15] This model is widely used to describe evaluations of educational interventions in a variety of contexts. Papers reporting more than one outcome level and active learning strategy were classified separately to calculate the number of evaluations per level and active learning strategy, respectively.

## Results

Searches in EMBASE (n = 1200), MEDLINE (n = 422), Scopus (n = 464), and ERIC (n = 132) databases generated 2,218 records. Duplicates (n = 808) and articles not published in English

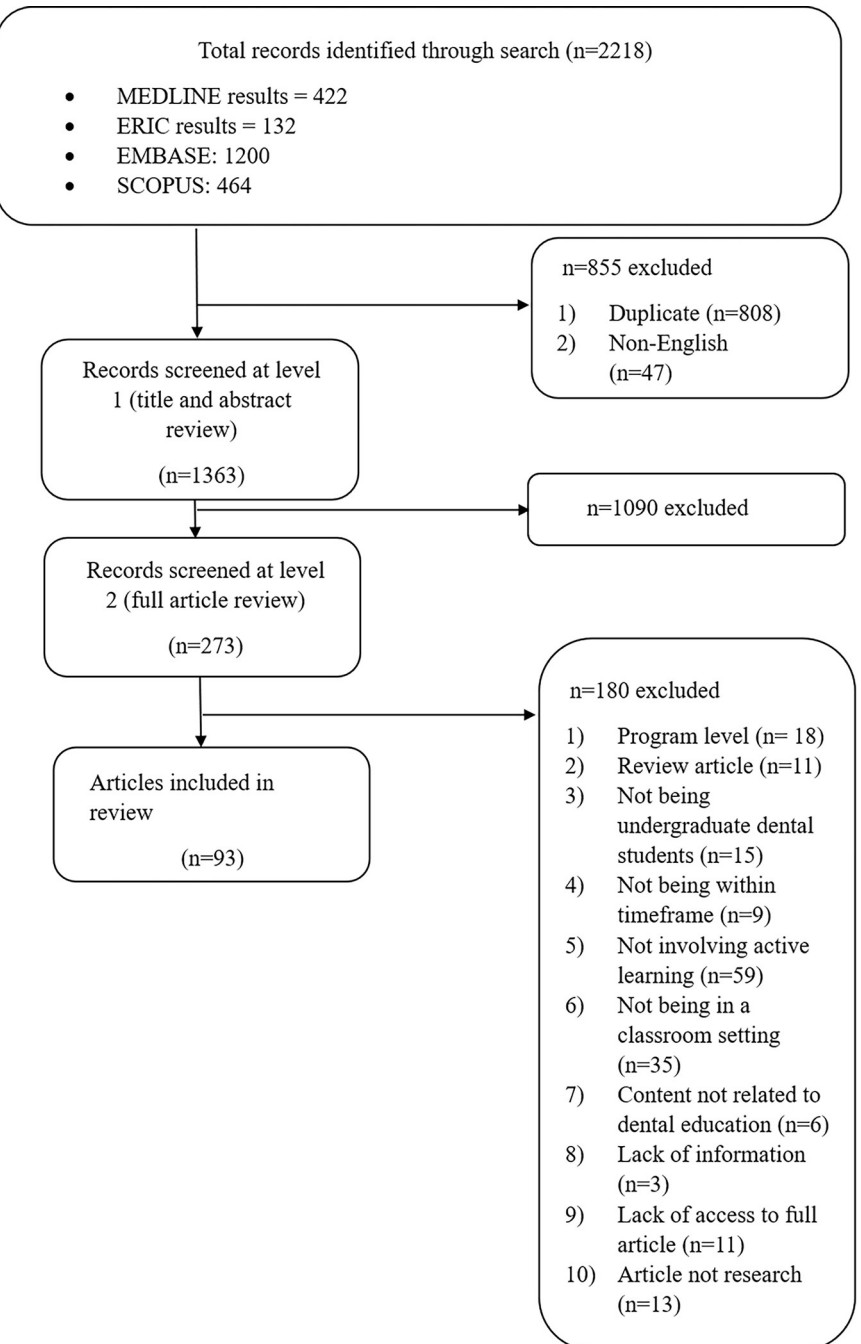

**Fig 1. Flow diagram of the study selection process.**

(n = 47) were removed. The screening of titles and abstracts yielded 273 potentially eligible articles and the screening of full texts identified 93 eligible articles, which were included in this review (Fig 1). No additional articles were identified through checking the reference lists of eligible studies and studies included in previous. A total of 10,473 students and 199 faculty were involved in the selected studies. Students involved were from dentistry (n = 10,297; 98.3%), medicine (n = 126; 1.2%), and dental hygiene (n = 50; 0.5%).

## Characteristics of reviewed studies

As shown in Table 2, selected studies originated from different geographical areas, including Asia (n = 46; 49.4%), North America (n = 29; 31.1%), Europe (n = 10; 10.7%), South America (n = 6; 6.4%), Australia (1) and Africa (1). Twenty-eight of the studies produced in North America were conducted in the United States and 1 in Canada. Thirty-one studies were published between 2005 and 2014 and 62 between 2015 and 2022. Nine studies (9.6%) did not indicate the content area. Most studies reported active learning in clinical (n = 54; 58%) and basic (n = 25; 26.8%) sciences, and only 5 (5.3%) in behavioral and social sciences.

Methodologically, most studies (n = 85; 91.3%) were quantitative in nature. Only a few used qualitative (n = 2; 2.1%) and mixed-method (n = 6; 6.4%) approaches. Most studies (n = 67; 72%) did not explicitly report the methodology used and some (n = 8; 8.6%) reported features of the methodology employed (e.g., prospective, comparative). Reported quantitative methods (n = 26; 27.9%) included pre- and post-tests (n = 6), randomized controlled trials (n = 4), cross-sectional studies (n = 3), cohort studies (n = 2), qualitative description (n = 1), case-control studies (n = 1), and experiments without randomization (n = 1). Two reported randomized controlled trials did not describe sequence generation, none reported allocation concealment details, and only 1 reported blinding of outcome assessors. Most common means of data collection included surveys (n = 74; 79.5%) and test scores (n = 59; 63.4%) alone or combined.

## Evaluation types and designs

All studies performed outcome evaluations. No process evaluations were reported alone or combined with outcome evaluations. Outcomes evaluated included satisfaction (n = 73), knowledge acquisition (n = 61), skill development (e.g., clinical, problem-solving, communication skills) (n = 3), and engagement (n = 2). Studies performed post-intervention (n = 70; 75.2%), pre-post-intervention (n = 18; 19.3%), and during-post-intervention (n = 5; 5.3%) evaluations.

Of all the evaluations performed (n = 93), post-intervention evaluations (n = 70) included a single group exposed to one condition (n = 23; 24.7%) or two compared conditions (n = 9; 9.6%), two compared groups exposed to two conditions including (n = 10; 10.7%) and not including (n = 21; 22.5%) randomization, and two or more non-compared groups exposed to one condition, including one-time (n = 6; 6.4%) or two-time (n = 1; 1.07%) evaluation points. In the one-time evaluation point, the outcome variables of interest were evaluated after the intervention, whereas in the two-time evaluation points, the outcome variables of interest were evaluated after the intervention by asking participants to assess those variables before and after the intervention. In both cases, the evaluation data of the study groups were aggregated. Pre-and-post intervention evaluations (n = 18), included a single group exposed to one condition (n = 4; 4.3%) or two compared conditions (n = 1; 1.07%), two compared groups exposed to two conditions including (n = 8; 8.6%), and not including (n = 4; 4.3%) randomization, and two or more non-compared groups exposed to one condition with one-time evaluation point (n = 1; 1.07%). During-post-intervention evaluations (n = 5), included a single group exposed to one condition (n = 1; 1.07%) or two compared conditions (n = 1; 1.07%) and two compared groups exposed to two conditions including (n = 1; 1.07%) and not including (n = 2; 2.1%) randomization.

## Evaluated active learning strategies

Studies evaluated several active learning strategies. Strategies frequently (more than 10 studies) and fairly (between 6 and 10 studies) evaluated included flipped learning, group discussion,

**Table 2. Summary of characteristics of reviewed studies.**

| Authors, year | Country | Inquiry | Study Design | Content Area | Active Learning Strategies | Comparator (if any) | Level of Evaluation |
|---|---|---|---|---|---|---|---|
| Mitchell & Brackett, 2017 [35] | USA | Quantitative | Not reported | Basic sciences | Flipped learning with TBL* | Traditional lecture | Reaction |
| Omar, 2017 [36] | Saudi Arabia | Quantitative | Not reported | Clinical sciences | Group discussions | N/A | Reaction |
| Gali et al., 2015 [37] | India | Quantitative | RCT** | Basic sciences | Group discussions | Traditional lecture | Reaction and Learning |
| Ihm et al., 2017 [38] | Korea | Quantitative | Not reported | Basic sciences | Flipped learning | Traditional lecture | Reaction |
| Kim et al., 2018 [39] | Korea | Quantitative | Not reported | Basic sciences | Flipped learning | Traditional lecture | Reaction and Learning |
| Luchi et al., 2017 [40] | Brazil | Quantitative | Not reported | Basic sciences | Game | Traditional lecture | Reaction and Learning |
| Almajed et al., 2016 [41] | Australia | Qualitative | Not reported | Not reported | Group discussion | Traditional lecture | Reaction |
| Ha-Ngoc & Park, 2015 [42] | USA | Quantitative | Not reported | Clinical sciences | Peer teaching | Traditional lecture | Reaction |
| Park et al., 2014 [43] | USA | Quantitative | Not reported | Clinical sciences | TBL*** | Individual learning | Learning |
| Miller et al., 2013 [44] | USA | Quantitative | Not reported | Basic sciences | Think-pair-share | Traditional lecture | Reaction and Learning |
| Khan, 2011 [45] | South Africa | Quantitative | Not reported | Clinical sciences | Group discussion | Active learning activities | Reaction |
| Kieser et al., 2008 [46] | New Zealand | Quantitative | Not reported | Clinical sciences | PBL | PBL | Reaction |
| Reich et al., 2007 [47] | Germany | Quantitative | Not reported | Clinical sciences | PBL | Traditional lecture | Reaction and Learning |
| Qutieshat et al., 2020 [48] | Jordan | Quantitative | Not reported | Clinical sciences | Flipped learning | Traditional lecture | Reaction and Learning |
| Ashwini et al., 2019 [49] | India | Quantitative | Not reported | Behavioral Sciences | Flipped learning | Traditional lecture | Reaction |
| Kohli et al., 2019 [50] | Malaysia | Quantitative | Cohort study | Clinical sciences | Flipped learning | Traditional lecture | Reaction and Learning |
| Tricio et al., 2019 [51] | Columbia | Mixed method | Not reported | Clinical sciences | Fishbowl | Traditional lecture | Reaction and Learning |
| Tauber et al., 2019 [52] | Czech Republic | Quantitative | Not reported | Basic sciences | Group discussion | Traditional lecture | Reaction and Learning |
| Himida et al., 2019 [53] | Scotland | Mixed method | Not reported | Behavioral sciences | Forum theatre | Traditional lecture | Reaction |
| Slaven et al., 2019 [54] | USA | Quantitative | Not reported | Clinical sciences | Flipped learning | Traditional lectures | Reaction and Learning |
| Park et al., 2019 [55] | USA | Quantitative | Not reported | Clinical sciences | TBL | Individual learning | Reaction and Learning |
| Yang et al., 2019 [56] | China | Quantitative | Not reported | Basic sciences | Group discussion | Traditional lectures | Reaction and Learning |
| Veeraiyan et al., 2019a [57] | India | Quantitative | Not reported | Basic sciences | TBL | Traditional lectures | Reaction and Learning |
| Veeraiyan et al., 2019b [58] | India | Quantitative | Retrospective | Clinical sciences | Flipped learning | Traditional lectures | Learning |
| Veeraiyan et al., 2019c [59] | India | Quantitative | Prospective | Clinical sciences | Flipped learning | Traditional lectures | Learning |
| Veeraiyan et al., 2019d [60] | India | Quantitative | Not reported | Clinical sciences | Flipped learning | Traditional lectures | Learning |
| Al-Madi et al., 2018 [61] | Saudi Arabia | Quantitative | Cross-sectional | Basic sciences | PBL | Traditional lectures | Reaction and Learning |

*(Continued)*

**Table 2.** (Continued)

| Authors, year | Country | Inquiry | Study Design | Content Area | Active Learning Strategies | Comparator (if any) | Level of Evaluation |
|---|---|---|---|---|---|---|---|
| Chutinan et al., 2018 [62] | USA | Mixed method | Not reported | Basic sciences | Flipped learning | Traditional lectures | Reaction and Learning |
| Jones, 2019 [63] | USA | Mixed method | Not reported | Clinical sciences | Group discussion | Traditional lectures | Reaction and Learning |
| Xiao et al., 2018 [64] | USA | Quantitative | Comparative | Basic sciences | Flipped learning | Traditional lectures | Reaction and Learning |
| Varthis & Anderson, 2018 [65] | USA | Quantitative | Not reported | Basic sciences | Blended learning | Traditional lectures | Reaction |
| Islam et al., 2018 [66] | Malaysia | Quantitative | Case control | Clinical sciences | Flipped learning | Traditional lectures | Reaction and Learning |
| Lee & Kim, 2018 [67] | USA | Quantitative | Not reported | Clinical sciences | Flipped learning | Traditional lectures | Reaction and Learning |
| Costa-Silva et al., 2018 [68] | Brazil | Quantitative | Not reported | Basic sciences | Group discussion | Traditional lectures | Learning |
| AbdelSalam et al., 2017 [69] | Saudi Arabia | Quantitative | Not reported | Basic sciences | Peer teaching | Traditional lectures | Learning |
| Bai et al., 2017 [70] | China | Mixed method | RCT | Clinical sciences | PBL | Traditional lectures | Reaction and Learning |
| Nishigawa et al., 2017a [71] | Japan | Quantitative | Cohort | Clinical sciences | TBL | Traditional lectures | Learning |
| Gadbury-Amyot et al., 2017 [72] | USA | Quantitative | Not reported | Clinical sciences | Flipped learning | Traditional lectures | Reaction |
| Sagsoz et al., 2017 [73] | Turkey | Quantitative | Pre- and post-test | Clinical sciences | Jigsaw method | Traditional lectures | Learning |
| Nishigawa et al., 2017b [74] | Japan | Quantitative | Not reported | Clinical sciences | Flipped learning | TBL | Learning |
| Samuelson et al., 2017 [75] | USA | Quantitative | Crossover | Clinical sciences | Group discussion | Traditional lectures | Reaction and Learning |
| Eachempati et al., 2016 [76] | Malaysia | Qualitative | Cross-sectional | Clinical sciences | Blended learning with group learning | Traditional lectures | Reaction |
| Cardozo et al., 2016 [77] | Brazil | Quantitative | Not reported | Basic sciences | Game | Traditional lectures | Learning |
| Bohaty et al., 2016 [78] | USA | Quantitative | Not reported | Clinical sciences | Flipped learning | Traditional lectures | Reaction |
| Echeto et al., 2015 [79] | USA | Quantitative | Not reported | Clinical sciences | TBL | Traditional lectures | Learning |
| Park & Howell, 2015 [80] | USA | Quantitative | Not reported | Basic Sciences | Flipped learning | Traditional lectures | Reaction |
| Takeuchi et al., 2015 [81] | Japan | Quantitative | Not reported | Clinical sciences | TBL | Traditional lectures | Reaction and Learning |
| Ilgüy et al., 2014 [82] | Turkey | Quantitative | Not reported | Clinical sciences | Group discussion | Traditional lectures | Learning |
| Guven et al., 2014 [83] | Turkey | Quantitative | Not reported | Basic sciences | PBL | Traditional lectures | Reaction and Learning |
| Du et al., 2013 [84] | China | Quantitative | Not reported | Clinical sciences | Group discussion | Traditional lectures | Reaction and Learning |
| Haj-Ali & Al Quran, 2013 [85] | United Arab Emirates | Quantitative | Not reported | Clinical sciences | TBL | Traditional lectures | Reaction and Learning |
| Ratzmann et al., 2013 [86] | Germany | Quantitative | Not reported | Clinical sciences | PBL | Traditional lectures | Reaction |
| McKenzie, 2013 [87] | USA | Quantitative | Pre-and post-test | Clinical sciences | Group discussion | Traditional lectures | Reaction |
| Kumar & Gadbury-Amyot, 2012 [88] | USA | Quantitative | Not reported | Clinical sciences | TBL | Traditional lectures | Reaction and Learning |

(*Continued*)

**Table 2.** (*Continued*)

| Authors, year | Country | Inquiry | Study Design | Content Area | Active Learning Strategies | Comparator (if any) | Level of Evaluation |
|---|---|---|---|---|---|---|---|
| Alcota et al., 2011 [89] | Chile | Quantitative | Not reported | Clinical sciences | PBL with debate and group discussion | Traditional lectures | Reaction and Learning |
| Romito & Eckert, 2011 [90] | USA | Quantitative | Not reported | Basic sciences | PBL | Traditional lecture | Learning |
| Obrez et al., 2011 [91] | USA | Quantitative | Not reported | Basic sciences | Group discussion | Traditional lecture | Reaction and Learning |
| Dantas et al., 2010 [92] | Brazil | Quantitative | Not reported | Clinical sciences | Group discussion | Text reading | Learning |
| Grady et al., 2009 [93] | UK | Quantitative | Not reported | Clinical sciences | Group discussion | Traditional lecture | Reaction |
| Moreno-López et al., 2009 [94] | Italy | Quantitative | Not reported | Clinical sciences | PBL | Traditional lecture | Learning and Reaction |
| Pileggi & O'Neill, 2008 [95] | USA | Quantitative | Not reported | Clinical sciences | TBL | Traditional lecture | Reaction and Learning |
| Park et al., 2007 [96] | USA | Quantitative | Retrospective | Clinical sciences | PBL with tutor expertise | PBL without tutor expertise | Learning and Reaction |
| Rich et al., 2005 [97] | USA | Quantitative | Not reported | Clinical sciences | PBL | Traditional lecture | Reaction |
| Croft et al., 2005 [98] | UK | Quantitative | Not reported | Behavioral sciences | Role Play | Traditional lecture | Reaction |
| Deepak et al., 2019 [58] | India | Quantitative | Prospective | Clinical sciences | Flipped learning | Traditional lecture | Learning |
| Qutieshat et al., 2018 [99] | Jordan | Quantitative | Not reported | Clinical sciences | Debate | Reply Speech | Reaction |
| Paul et al., 2019 [100] | Malaysia | Quantitative | Cross-sectional | Clinical sciences | Blended learning | Traditional lecture | Reaction and Learning |
| Youssef et al., 2012 [101] | Egypt | Quantitative | Not reported | Basic sciences | Group discussion | Traditional lecture | Reaction |
| Al Kawas & Hamdy, 2017 [102] | United Arab Emirates | Mixed method | Not reported | Not reported | TBL | Traditional lecture | Reaction |
| Nishigawa et al., 2017c [74] | Japan | Quantitative | Not reported | Clinical sciences | TBL and flipped learning | Flipped learning | Reaction and Learning |
| Khan et al., 2012 [103] | Malaysia | Quantitative | Not reported | Basic sciences | Debate | Traditional lecture | Reaction |
| Katsuragi, 2005 [104] | Japan | Quantitative | Not reported | Basic sciences | PBL | Traditional lecture | Reaction and Learning |
| Zhang et al., 2012 [105] | China | Quantitative | Not reported | Clinical sciences | PBL | Traditional lectures | Reaction and Learning |
| Zain-Alabdeen, 2017 [106] | Saudi Arabia | Quantitative | Not reported | Clinical sciences | Flipped learning | Traditional lectures | Reaction |
| Elledge et al., 2018 [107] | UK | Quantitative | Not reported | Clinical sciences | Flipped learning | Traditional lectures | Reaction and Learning |
| Richards & Inglehart, 2006 [108] | USA | Quantitative | Not reported | Clinical sciences | Group discussion | Traditional lectures | Reaction |
| Tack & Plasschaert, 2006 [109] | Netherlands | Quantitative | Not reported | Clinical sciences | PBL | Traditional lectures | Reaction and Learning |
| Markose et al., 2018 [110] | India | Quantitative | Comparative | Behavioral sciences | PBL | Traditional lectures | Reaction and Learning |
| Ahmadian et al., 2017 [111] | Iran | Quantitative | Interventional | Behavioral sciences | PBL | Role play | Reaction |
| Metz et al., 2015 [112] | USA | Quantitative | Not reported | Clinical sciences | Group discussion | Traditional lectures | Reaction and Learning |
| Shigli et al., 2017 [113] | India | Quantitative | Experiment | Clinical sciences | Group discussion | Traditional lectures | Reaction |

(*Continued*)

**Table 2.** (Continued)

| Authors, year | Country | Inquiry | Study Design | Content Area | Active Learning Strategies | Comparator (if any) | Level of Evaluation |
|---|---|---|---|---|---|---|---|
| Roopa et al., 2013 [114] | India | Quantitative | Not reported | Basic sciences | Peer teaching | Traditional lectures | Reaction |
| Rimal et al., 2015 [115] | Nepal | Quantitative | Not reported | Basic sciences | PBL | Traditional lectures | Reaction |
| Ihm et al., 2017 [116] | Republic of Korea | Quantitative | Not reported | Not reported | PBL | Traditional lectures | Learning |
| Chandelkar & Kulkarni, 2014 [117] | India | Quantitative | Not reported | Basic sciences | Peer teaching | Traditional lectures | Reaction and Learning |
| Huynh et al., 2022 [118] | USA | Quantitative | Not reported | Clinical sciences | Blended Learning | Traditional lectures | Reaction |
| Özcan, 2022 [119] | USA | Quantitative | Not reported | Clinical sciences | Flipped Learning | Traditional lectures | Learning and Reaction |
| Gallardo et al., 2022 [120] | Spain | Quantitative | Pre- and post-test | Clinical sciences | Flipped Learning | Traditional lectures | Reaction |
| Alharbi et al., 2022 [121] | Saudi Arabia | Quantitative | Pre- and post-test | Not reported | Flipped Learning | Traditional lectures | Learning and Reaction |
| Zhou et al., 2022 [122] | China | Quantitative | Pre- and post-test | Clinical sciences | Flipped Learning | Traditional lectures | Reaction |
| Xiao et al., 2021 [123] | USA | Quantitative | Pre- and post-test | Basic science | Flipped Learning | Traditional lectures | Learning |
| Veeraiyan et al., 2022 [124] | India | Quantitative | Not reported | Not reported | Multiple active learning strategies | NA | Learning |
| Ganatra et al., 2021 [125] | Canada | Mixed method | Not reported | Clinical sciences | Think pair share | NA | Reaction |

*TBL: Team-based learning

**RCT: Randomized control trial

***PBL: Problem-based learning

problem-based learning (PBL), and team-based learning (TBL). Blended learning, peer teaching, debate, and role play were occasionally evaluated (between 3 to 5 studies). Strategies seldom evaluated (1 or 2 studies) included games, think-pair-share, and others such as fishbowl and Jigsaw. All outcome evaluations were performed at reaction and learning levels as the present review focused on classroom dental education. Thirty-two studies (34.4%) performed reaction evaluations alone, 19 (20.4%) learning evaluations alone, and 42 (45.1%) reaction and learning evaluations combined. Only 4 studies (4.3%) reported faculty data in addition to student data. The lengths of the exposures to active learning ranged from one hour to three years.

## Reaction-level evaluations, including self-reported learning

Seventy-six student reaction evaluations alone or combined were conducted. In these evaluations, active learning was perceived to improve satisfaction in 66 studies (86.8%) and knowledge acquisition in 4 studies (5.3%). Sixty-five of these evaluations or studies compared active learning and lectures, 3 compared two active learning strategies, and 3 compared different forms of the same active learning strategy. In fifty-nine studies, active learning was perceived as superior to lectures, 5 found no differences between active learning and lectures, and only 1 reported lectures as superior to active learning. Only 4 evaluations reported instructors' reaction data. In all these evaluations, instructors positively valued active learning.

Frequently, fairly, and occasionally evaluated (three or more studies) strategies using reaction-level data included flipped learning, PBL, group discussion, TBL, and blended learning. Peer teaching, role play debate, game, and think-pair-share were seldom evaluated (1 or 2

studies) using reaction data. Flipped learning was perceived to improve satisfaction in 16 studies and was regarded as superior to lectures in 16 studies. PBL was viewed as effective to improve knowledge acquisition in 2 studies and satisfaction in 13 studies and perceived as superior to lectures in 13 studies. Group discussion was deemed effective for knowledge acquisition in 1 study and satisfaction in 12 studies and reported to be superior to lectures in 12 studies. TBL was viewed as beneficial to improve knowledge in 1 study and satisfaction in 7 studies and considered more effective than lectures in 7 studies. Blended learning was deemed to improve satisfaction in 4 studies and regarded as superior to lectures in 4 studies.

### Learning-level evaluations

All studies in which learning was directly measured (n = 57) found that active learning was effective to improve knowledge acquisition largely based on test scores. Forty-eight of these studies (84.2%) compared active learning and lectures and 4 studies (7.0%) compared two active learning strategies. Based on the learning data, 39 studies found that active learning was superior to lectures in knowledge acquisition and 9 reported no difference between active learning and traditional lectures.

Frequently and fairly evaluated strategies using direct measures of learning included flipped learning, PBL, group discussion, and TBL. Blended learning, peer teaching, debate, game, and think-pair-share were rarely evaluated using such measures. Based on direct learning data, flipped learning was found to improve knowledge acquisition in 12 studies and to be more effective than lectures in knowledge acquisition in all 12 studies. Similarly, PBL was found to enhance knowledge acquisition in 9 studies and to be superior to traditional teaching in knowledge acquisition in all 9 studies. Direct learning data also supported the effectiveness of group discussion and TBL. Specifically, group discussion and TBL were found to improve knowledge acquisition in 5 and 7 studies, respectively. Regarding this outcome, group discussion was reported to be more effective than lectures in 5 studies and TBL in 7 studies.

### Discussion

Most studies on active learning in classroom dental education were quantitative in nature and published in the last decade, did not report the study methodology, performed outcome evaluations, engaged in post-intervention evaluations, relied on student data, mainly measured satisfaction and knowledge acquisition, and focused on clinical and basic sciences. Our review also revealed that flipped learning, group discussion, problem-based learning, and team-based learning were the active learning strategies most frequently evaluated in classroom dental education. Based on both reaction and factual (direct measure) data, these strategies improved satisfaction and knowledge acquisition and were superior to traditional lectures in improving these outcomes. To our knowledge, this is the first attempt to map the literature on active learning strategies in classroom dental education. Our findings provide a much-needed overview of this body of literature, which previous strategy-specific reviews were not in a position to provide [10, 16, 17]. Such an overview is of critical importance to describe the available evidence and inform future research directions on the study topic.

Consistent with the data from previous reviews, the number of studies on active learning in dental education has increased over time, especially within the last decade [9, 16, 18]. This shows a positive response to repeated calls for transforming the learning environments in dental education. This surge of publications is encouraging as a proxy for innovation in dental education and as a vehicle for knowledge dissemination among dental researchers and educators. In research, though, more publication does not necessarily mean better research activity. Although scoping reviews are not intended to assess the quality of the studies conducted and

the credibility of the evidence generated [13], they can shed light on these issues based on the research methods and designs employed and the nature of the evidence produced. Quality of research in educational innovations can also be inferred by examining the types of evaluations conducted.

Most studies included in our review did not explicitly indicate the methodology used, which previous review research in medical education has also reported [19]. This is of concern as methodologies are supposed to be deliberately chosen to inform study designs [20]. We did not assess whether the reported methodologies were correctly classified; however, misclassifications of study methodologies have been documented [21, 22]. Such misclassifications may be due to lack of methodological understanding and attempts to pursue methodological credibility by claiming the use of "more robust" designs than those actually employed [22]. Several recommendations have been made to help researchers frame their projects methodologically and conceptually, including the engagement of methodologists throughout the research process [19].

Many studies included in our review employed a post-intervention evaluation design with a single cohort. This design is known to have several limitations, such as the inability to assess the magnitude of the improvement, if any, and to account for extraneous variables that may influence the learning outcomes apart from the intervention. Additionally, none of the studies included in our review reported process evaluation. This type of evaluation examines the extent to which an intervention was implemented as expected, met the parameters of effectiveness for the intervention (conditions under which it works), and was aligned with the underlying principles of the type of learning (e.g., collaborative learning) it aimed to promote [23]. Process evaluations are particularly helpful to determine whether an intervention did not work because of its effectiveness, implementation, or both. Failure to report process evaluation and properly design and implement active learning strategies have been previously documented [6]. Such shortcomings can be misleading in two fundamental ways: suggesting that a strategy was not effective when it could potentially be and suggesting it was delivered as expected when it was not.

Our findings highlight the importance of reporting not only the research inquiries (e.g., quantitative, qualitative) and methodologies (e.g., cross-sectional, RCT), but also the specific evaluation designs employed in the studies. Since methododologies may not be reported or properly classified, the specific evaluation design used becomes the best proxy for the quality of the outcome evaluation performed. This aspect should be determined by the researchers conducting the review because it may not be clearly defined in published papers. Our classification of evaluation designs can be used for this purpose, although further research may be needed to ascertain its instrumental value.

Few studies in our review used qualitative and mixed-method designs, which best practices in curriculum evaluation at the course and program levels recommend [24]. Such practices include using multiple evaluators, collecting and combining qualitative and quantitative data to provide a comprehensive evaluation, and using an evaluation framework (e.g., a logic model) to guide the evaluation process. Qualitative research is particularly suited to shed light into the circumstances under which interventions work (why and how) and the contextual factors shaping the outcomes of interventions and participants' experiences [25].

Reviews on active learning in dental and medical education have revealed that active learning strategies are commonly evaluated using student feedback [6, 9]. Our study confirms the use of student feedback as the main source of evaluation, which is useful to judge some aspects of teaching effectiveness, such as engagement and organization, but not others such as appropriateness of the pedagogical strategy used to achieve the learning objectives [26]. Faculty feedback is important to comprehensively evaluate active learning across health professions

education and ascertain their uptake and continued use of active learning strategies in classroom and clinical learning environments.

Similar to previous review research on active learning across health professions education [5], many studies included in our review used reactionary and factual data to evaluate the impact of several active learning strategies on the outcomes of interest, especially knowledge acquisition. This is an important strength of the literature on active learning in classroom dental education. Reaction- and learning-level outcome evaluations serve slightly different purposes, but both are needed to establish whether students and faculty are satisfied with the active learning strategies used and the actual impact of those strategies on knowledge acquisition, skill development, and attitudinal change. Further research is needed to critically appraise the validity of the means used to collect direct measures of learning, especially when knowledge tests were not originally developed and validated for research purposes.

Satisfaction and knowledge acquisition were the main outcomes evaluated in the studies included, while skill development (e.g., critical thinking, problem-solving skills) was infrequently considered. The latter is an important learning outcome in the context of competency-based education, which has been highly recommended in dental education [27]. Failure to measure whether active learning promotes important skills in this type of education may be due to the length and nature of the exposures (interventions) needed to achieve these outcomes and "inherent" difficulties to measure high-level outcomes [28].

Research on active learning in classroom dental education reflects the emphasis that traditional dental programs place on basic and clinical sciences. We identified only a few papers on active learning in behavioral and social sciences, which are a key component of dental education. These sciences have expanded the understanding of diseases beyond their biological determinants and that of treatment and management beyond clinical procedures [29]. Additionally, behavioral sciences provide dental students with competencies for personalized care, inter-professional care, disease prevention and management, and personal well-being of patients and care providers [30]. While integrating the behavioral science curriculum remains an important task [31], our findings suggest that research is warranted to demonstrate the effectiveness of active learning in delivering behavioral science content in dental education.

Active learning strategies most frequently evaluated in classroom dental education (flipped learning, group discussion, PBL, TBL) are similar to those commonly evaluated in dental and medical education [5, 9, 32]. Properly evaluated strategies provide dental educators with a menu of teaching options from which to choose the most suitable strategy(ies) to achieve their learning objectives. However, other active learning strategies (e.g., peer teaching, role play, think-pair-share) need to be further evaluated in dental education as they have proven effective to achieve certain learning objectives alone or in conjunction with other strategies [33, 34].

Despite the diversity of research designs, populations, settings, and evaluated strategies, active learning in classroom dental education was positively valued by students and faculty, was perceived as beneficial and 'proven' effective to promote satisfaction and knowledge acquisition, and was found to be superior to traditional lectures to promote these outcomes. These findings are consistent with those of previous reviews in dental education and health professions education in general [6, 9, 16]. Given the limitations of traditional lectures to promote deep and meaningful learning, dental researchers are encouraged to compare active learning strategies to achieve similar generic and specific learning objectives in order to demonstrate their relative effectiveness to achieve those objectives.

Our review also uncovered several reporting issues. These issues included not reporting or underreporting the research methodology, key aspects (e.g., allocation concealment) of the research design, characteristics of the instruments used for data collection, validity evidence of those instruments, active learning strategies employed, and length of the exposure to those

strategies. Sufficient details of studies' designs and conduct are important to judge the quality of the studies and that of the evidence produced. For example, without knowing the actual length of the exposure, it is not possible to appraise whether the expected learning outcomes were not achieved because the strategy used was not effective or because the exposure to the strategy was insufficient.

Limitations of our study encompasses general limitations of scoping reviews and study-specific limitations. General limitations include the potential for publication bias (published literature often prioritizes studies with significant findings over those with non-significant findings) and the absence of quality assessments for the included studies. While this assessment is not required in scoping reviews, it is important to note that the research designs of most included studies do not offer sufficient evidence to demonstrate the effectiveness of active learning in dental education classrooms. Several study-specific limitations need to be acknowledged. We relied on authors' classifications of research methodologies and active learning strategies, which may not be the actual methods and strategies used. Misclassification of active learning strategies has been previously reported [17]. We excluded papers in languages other than English due to limited resources for translation, which may impact the generalizability of our study findings. However, based on the number of papers included, we are confident that the inclusion of this literature would not have changed the patterns observed in the extracted data. Our summary of the main results of previous studies by level of outcome evaluation (reaction and learning) may not account for noticeable differences in study design, sample, settings, and measures across studies.

## Conclusion

Although active learning strategies were positively valued and found effective using reaction and factual data, robust evaluation designs are needed to further demonstrate their effectiveness in classroom dental education. Aside from effectiveness questions, other issues remain to be elucidated, including for whom, how, when, and in what respect active learning may work in dental education. Future research should evaluate not only the impact of active learning strategies on satisfaction and knowledge acquisition, but also on skill development to support competency-based teaching and assessment in dental education. Similarly, active learning should be used and evaluated across all the main components of dental education, including behavioral and social sciences. Dental education journals should encourage researchers to comply with evaluation and reporting standards for educational innovations to ensure that these innovations are designed, conducted, and reported as expected.

## Supporting information

**S1 Checklist. Preferred Reporting Items for Systematic reviews and Meta-Analyses extension for Scoping Reviews (PRISMA-ScR) checklist.**
(DOCX)

## Acknowledgments

The authors would like to thank Drs. Tania Doblanko and Tanushi Ambekar for their involvement in the study design and preliminary search for relevant articles.

## Author Contributions

**Conceptualization:** Arnaldo Perez, Maryam Kebbe, Seema Ganatra, Geoff Ball.

**Data curation:** Mohammad Moharrami, Silvia Gianoni-Capenakas, Seema Ganatra, Nazlee Sharmin.

**Formal analysis:** Arnaldo Perez, Jacqueline Green, Nazlee Sharmin.

**Funding acquisition:** Arnaldo Perez, Jacqueline Green.

**Investigation:** Arnaldo Perez, Jacqueline Green, Mohammad Moharrami, Maryam Kebbe, Seema Ganatra, Geoff Ball, Nazlee Sharmin.

**Methodology:** Arnaldo Perez, Silvia Gianoni-Capenakas, Maryam Kebbe, Geoff Ball, Nazlee Sharmin.

**Project administration:** Seema Ganatra.

**Supervision:** Arnaldo Perez.

**Writing – original draft:** Arnaldo Perez, Jacqueline Green.

**Writing – review & editing:** Arnaldo Perez, Jacqueline Green, Mohammad Moharrami, Silvia Gianoni-Capenakas, Maryam Kebbe, Seema Ganatra, Geoff Ball, Nazlee Sharmin.

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
