## [Decision Letter · Decision Letter 0]

20 Sep 2023

PONE-D-23-12023Active learning in undergraduate classroom dental education: a scoping reviewPLOS ONE

Dear Dr. Perez,

Thank you for submitting your manuscript to PLOS ONE. After careful consideration, we feel that it has merit but does not fully meet PLOS ONE’s publication criteria as it currently stands. Therefore, we invite you to submit a revised version of the manuscript that addresses the points raised during the review process. The reviewers have expressed their satisfaction with the paper and supported its acceptance after fixing some details. More importantly, you need to highlight the limitations of the paper, the limitation of scoping reviews and make sure that every recommendation is based on actual data in your review. 

We look forward to receiving your revised manuscript.

Kind regards,

Mohammed Saqr, Ph.D

Academic Editor

PLOS ONE

Journal Requirements:

Reviewers' comments:

Reviewer's Responses to Questions

**Comments to the Author**

1. Is the manuscript technically sound, and do the data support the conclusions?

Reviewer #1: Yes

Reviewer #2: Yes

Reviewer #3: Yes

2. Has the statistical analysis been performed appropriately and rigorously? 

Reviewer #1: N/A

Reviewer #2: N/A

Reviewer #3: I Don't Know

3. Have the authors made all data underlying the findings in their manuscript fully available?

Reviewer #1: Yes

Reviewer #2: Yes

Reviewer #3: Yes

4. Is the manuscript presented in an intelligible fashion and written in standard English?

Reviewer #1: Yes

Reviewer #2: Yes

Reviewer #3: Yes

5. Review Comments to the Author

Reviewer #1: Excellent review.....Educational research is sorely lacking in the health professions, and this scoping review is an important contribution. My only comment would be to address any potential effect of not including

non-English papers in the discussion section.

Reviewer #2: Informative study, well written.

Author followed PRISMA guidelines for scoping reviews, and to my knowledge followed the journal guidelines.

Typos:

Line 177: Methodologically

I, myself learned more on the subject.

Reviewer #3: the paper is very will written with thorough explanation of finding and interpretations. The topic is very well selected since there are limited reviews on it. The choice of a scoping review is successful since to gives a different depth of information.

6. PLOS authors have the option to publish the peer review history of their article (what does this mean?). If published, this will include your full peer review and any attached files.

Reviewer #1: **Yes: **Elliot Abt

Reviewer #2: No

Reviewer #3: No

---

## [Author Response · Author response to Decision Letter 0]

25 Sep 2023

Reviewer and editor comments were included in a separate file

---

## [Editor Report · Decision Letter 1]

9 Oct 2023

Active learning in undergraduate classroom dental education: a scoping review

PONE-D-23-12023R1

Dear Dr. Perez,

We’re pleased to inform you that your manuscript has been judged scientifically suitable for publication and will be formally accepted for publication once it meets all outstanding technical requirements.

Kind regards,

Mohammed Saqr, Ph.D

Academic Editor

PLOS ONE
---

## [Editor Report · Acceptance letter]

18 Oct 2023

PONE-D-23-12023R1 

Active learning in undergraduate classroom dental education- a scoping review 

Dear Dr. Perez:

I'm pleased to inform you that your manuscript has been deemed suitable for publication in PLOS ONE. Congratulations! Your manuscript is now with our production department. 

Kind regards, 

on behalf of

Dr. Mohammed Saqr 

Academic Editor

PLOS ONE